# Improvement in the Microbial Resistance of Resin-Based Dental Sealant by Sulfobetaine Methacrylate Incorporation

**DOI:** 10.3390/polym12081716

**Published:** 2020-07-30

**Authors:** Myung-Jin Lee, Utkarsh Mangal, Se-Jin Kim, Yeo-Phil Yoon, Eun-So Ahn, Ee-Seul Jang, Jae-Sung Kwon, Sung-Hwan Choi

**Affiliations:** 1Department of Dental Hygiene, Division of Health Science, Baekseok University, Cheonan 31065, Korea; dh.mjlee@gmail.com; 2Department of Orthodontics, Yonsei University College of Dentistry, Seoul 03722, Korea; utkmangal@yuhs.ac; 3Department of Dental Hygiene, College of Medical Science, Konyang University, Daejeon 32992, Korea; tpwlsdl19@naver.com (S.-J.K.); yeophil@naver.com (Y.-P.Y.); dksdmsth@nate.com (E.-S.A.); dltmfdl1030@naver.com (E.-S.J.); 4Department and Research Institute of Dental Biomaterials and Bioengineering, Yonsei University College of Dentistry, Seoul 03722, Korea; 5BK21 PLUS Project, Yonsei University College of Dentistry, Seoul 03722, Korea

**Keywords:** resin-based sealant, zwitterion, oral bacteria, bacterial adhesion, protein adsorption

## Abstract

Prevention of dental caries is a key research area, and improvement of the pit and fissure sealants used for caries prevention has been of particular interest. This report describes results of incorporating a zwitterion, sulfobetaine methacrylate (SB), into photo-polymerized resin-based sealants to enhance resistance to cariogenic bacteria and protein adhesion. Varying amounts (1.5–5 wt%) of SB were incorporated into a resin-based sealant, and the flexural strength, wettability, depth of cure, protein adhesion, bacterial viability, and cell cytotoxicity of the resultant sealants were evaluated. The flexural strength decreased with the increasing SB content, but this decrease was statistically significant only for sealants containing ≥3 wt% SB. Incorporating a zwitterion led to a significant reduction in the water contact angle and protein adhesion. The colony-forming unit count showed a significant reduction in the bacterial viability of *S. mutans*, which was confirmed with microscopic imaging. Moreover, cell cytotoxicity analysis of SB-modified sealants using an L929 fibroblast showed a cytotoxicity comparable to that of an unmodified control, suggesting no adverse effects on the cellular metabolism upon SB introduction. Hence, we conclude that the addition of 1.5–3 wt% SB can significantly enhance the inherent ability of sealants to resist *S. mutans* adhesion and prevent dental caries.

## 1. Introduction

Dental caries is a chronic oral ailment that has caused a worldwide oral health burden [1]. It is caused by an interplay of specific acidogenic bacteria, such as *Streptococcus mutans*, which are present in dental plaque surrounding external tooth surfaces. The interaction of such microbes with the tooth structure in the presence of carbohydrates leads to the initiation of irreversible damage in the form of dental caries [2]. Dental caries impacts people of all ages, and thus, it demands effective preventive measures. Given the multifactorial nature of dental caries, tooth morphology with deep pits and fissures is a risk factor because it presents potential sites for caries formation.

Various materials such as toothpaste, mouth rinse and fluoride varnish have been used to prevent dental caries. Since the pits and fissures of the occlusal surface have complicated and irregular structures, it is difficult to prevent dental caries using these materials [1]. To address the risk of caries formation due to pits and fissures, the adoption of preventive measures within the first few years of tooth eruption is commonly advised [2,3]. The preferred preventative measure is the use of dental resin-based sealants, which are designed to develop a physical barrier that prevents biofilm growth by blocking nutrition. Sealants are applied for the management of both the initial occlusal and proximal surface lesions [4]. Thus, researchers have explored several modifications to improve the properties of existing materials, focusing on remineralization and fluoride release [5,6].

The primary role of dental sealants is to act as a barrier to tooth–microbe interaction. Bacterial attachment can also be prevented by coating the tooth with hydrophilic materials. In this regard, recent research on sealants has focused on altering the surface ionic interactions to impart antifouling activity [7,8,9]. This has been achieved by incorporating zwitterions, compounds which contain both cationic and anionic moieties. The most common cationic moiety is a quaternary ammonium group, while various anionic moieties are also possible [7]. On the surface of the dental material containing zwitterions, a thick hydration layer is formed to block the adhesion of the salivary protein, thereby preventing oral bacteria and fungi from attaching to the surface of the material [10,11]. In particular, the synergetic effect between the bioactive components of resin-based material was evident by an increased hydrophilicity from the addition of zwitterion. A previous study showed antimicrobial properties as well as acid-neutralizing capacities due to the zwitterionic nature of the components [11]. Generally, acid produced by carbohydrate-fermenting acidogenic bacteria and demineralization dominate when the pH is below the critical pH of 5.5, which results in the dissolution of the enamel minerals. According to a previous study [10,11], the hydrophilicity of zwitterions allowed water to easily reach the core of the base material, which increases the ion release of the bioactive dental material, which promotes the acid-neutralizing effect.

Among zwitterionic compounds with antifouling properties, sulfobetaine methacrylate (SB) has often been studied due to its ease of production and use [12]. Diverse biomedical applications of SB as a non-fouling agent in sensors and membrane coatings have been reported [13,14]. Moreover, dental cements and resin-based composites containing zwitterions, including methacryloyloxyethyl phosphorylcholine (MPC) and dimethylaminohexadecyl methacrylate (DMAHDM), have recently been reported. These studies have indicated that the modified resin moieties can impart an improved bacterial resistance [8,15,16]. However, the incorporation of SB into the resin base as a preventive therapy agent has rarely been reported [10].

Accordingly, the aim of this study was to evaluate the mechanical properties, antimicrobial activity, and cytotoxicity of low percentages of SB-incorporated resin-based sealant. The null hypothesis was that the incorporation of SB will have no significant effect on the mechanical properties, bacterial resistance, or cytotoxicity of the resin-based sealant.

## 2. Materials and Methods

### 2.1. Incorporation of SB into Sealant

Sulfobetaine methacrylate (Sigma-Aldrich, St. Louis, MO, USA) was used in the study. SB powder was mixed into a sealant (Clinpro™ Sealant; 3 M ESPE, St. Paul, MN, USA) at 1.5%, 3%, and 5% by weight (wt%). An unmodified sealant without the incorporation of SB was used as a control. From the results of preliminary experiments, it was confirmed that the content of SB at 5 wt% or more could dramatically decrease the mechanical properties of the base material, and based on this, the composition of the test group was confirmed in this study. Four sealant compositions were tested, as listed in Table 1. In each experiment, at least five specimens were tested for each group.

### 2.2. Mechanical Properties

Mechanical properties were measured according to ISO 4049 [11]. For each group in Table 1, the bar-shaped samples (25 mm × 25 mm × 2 mm) were prepared. Samples were stored in distilled water at 37 ± 1 °C for 24 h after fabrication. A computer-controlled universal testing machine (Model 3366; Instron^®^, Norwood, MA, USA) was used to fracture the specimens in a three-point flexure. The flexural strength (σ) was calculated as
σ = 3Fl/(2*bh*^2^)(1)
where F is the maximum load on the specimen (in N), l is the distance between the two supports (20 mm), *b* is the width (in mm) of the specimen measured before the test, and *h* is the thickness (in mm) of the specimen measured immediately before the test.

### 2.3. Wettability

Wettability was determined in accordance with previous studies, using a video contact angle goniometer (SmartDrop; Femtobiomed Inc., Gyeonggi-do, Korea) [11] with distilled water as the reference liquid. Four groups of samples were fabricated in a mold with a diameter of 15 mm and a thickness of 1 mm. A total of 2 µL of distilled water was placed on the sample surface, and the contact angle with the sample surface was measured after 10 s.

### 2.4. Depth of Cure

According to ISO 4049, sealant samples were placed in a cylindrical Teflon mold, having a length of 10 mm and a diameter of 4 mm. A polyester film with a glass slide was then pressed over the mold, taking care to not generate bubbles. After confirming that the sealant material had completely filled the mold, the sample was irradiated for 20 s with an irradiation source directly above the sample. After polymerization, the sample was separated from the mold and immediately removed from the unpolymerized material using a plastic spatula. The height of the polymerized material was then measured with an electronic vernier caliper (Mitutoyo Corporation, Kawasaki, Kanagawa, Japan). Five specimens for each group in Table 1 were prepared and evaluated.

### 2.5. Protein Adsorption

Protein adsorption was tested following a previously established method [10,11]. Specimens were immersed into fresh phosphate-buffered saline (PBS; Gibco, Grand Island, NY, USA) for 1 h at room temperature, followed by immersion into a protein solution of bovine serum albumin (BSA; Difco, Sparks, MD, USA) and a bovine heart infusion (BHI; Difco, Sparks, MD, USA) broth (100 μL; 2 mg/mL) in PBS. After an incubation at 37 °C for 1 h, the specimens were gently rinsed twice with fresh PBS. After 4 h of incubation under sterile humid conditions (37 °C in 5% CO_2_), any unadhered protein was removed by washing twice with PBS. The amount of adsorbed protein was measured using 200 μL of bicinchoninic acid (Pierce Biotechnology Inc., Rockford, IL, USA), followed by an incubation at 37 °C for 30 min. A quantitative analysis of adsorbed proteins on the specimen surface was performed using a Micro BCA™ Protein Assay Kit (Pierce Biotechnology, Rockford, IL, USA). The optical density (OD) of each sample was measured using a microplate reader (Epoch; BioTek Instruments, Winooski, VT, USA) at 562 nm.

### 2.6. Colony-Forming Units

The strain used in the experiments was *Streptococcus mutans* (*S. mutans*, ATCC 25175). *S. mutans* was cultured in a BHI broth at 37 °C for 24 h. After preparing disk-shaped sealant samples, 1 mL of the bacterial suspension (1 × 10^6^ cells/mL) was placed on each disk in a 24-well plate and incubated at 37 °C for 24 h in a >95% humidity atmosphere. After incubation, the samples were gently washed twice with PBS to remove any non-adherent bacteria. The number of bacterial colony-forming units (CFUs) were counted according to the procedures reported in earlier studies. The adherent bacteria were harvested in 1 mL of PBS by sonication (SH−2100; Saehan Ultrasonic, Seoul, Korea) for 5 min [10,11], and 100 µL of this bacterial suspension was spread onto a BHI agar plate and incubated at 37 °C for 24 h. The total number of colonies was then counted.

### 2.7. Bacterial Viability

The viability of the adhered bacteria was determined by staining using a LIVE/DEAD bacterial viability kit (Molecular Probes, Eugene, OR, USA) according to the manufacturer’s protocols [10,11]. Equal volumes of SYTO 9 dye and propidium iodide (which stain live and dead bacteria, respectively) from the kit were mixed thoroughly. Subsequently, 3 µL of the mixture was added to 1 mL of the bacterial suspensions prepared as described above. After 15 min of incubation at room temperature in the dark, the stained samples were examined by confocal laser microscopy (CLSM, LSM880; Carl Zeiss, Thornwood, NY, USA). Live bacteria appeared green, while dead bacteria appeared red.

### 2.8. Cell Cytotoxicity

Tests on the extracts described below were carried out in accordance with ISO 10993-12, as established in an earlier study [17]. Briefly, culture medium (RPMI 1640; Gibco, Grand Island, NY, USA) was added to each bottle at a ratio of 0.2 g per mL of the sample. L929 cells (mouse fibroblast, NCTC clone 929, Korean Cell Line Bank, Korea) were seeded in 100 mL of culture medium at a density of 1 × 10^5^ cells/mL in 96-well plates (SPL, Pocheon-Si, Gyeonggi-Do, Korea) for 24 h. The culture medium was removed from the wells, and extractions from the samples or serial dilutions of extractions using culture medium (50%, 25%, 12.5%, and 6.25%) were placed into each well. These extracts were used for the cytotoxicity test in the 3-(4, 5-dimethyl thiazol-2-yl)-2, 5-diphenyl tetra zolium bromide (MTT) assay. The positive control was 1% phenol solution, and the negative controls were extracts of aluminum oxide ceramic. Extractions and dilutions of extractions were left for 24 h, after which the culture medium was removed and replaced with 50 µL of MTT solution (Sigma, St. Louis, MO, USA). After 2 h, the MTT solution was discarded, and 100 µL of isopropanol (Sigma, St. Louis, MO, USA) was added to each well. The plates were shaken until all the crystals were dissolved. The absorbance was spectrophotometrically measured using an ELISA reader (Epoch, BioTek, Winooski, VT, USA) at 570 nm.

### 2.9. Statistical Analysis

All statistical analyses were performed using IBM SPSS software, version 23.0 (IBM Korea Inc., Seoul, Korea) for Windows. The data are expressed as the mean ± S.D. of at least three independent experiments. The statistical significance was evaluated by a one-way analysis of variance with Tukey’s post hoc test. Values of *p* < 0.05 were considered statistically significant.

## 3. Results and Discussion

In this study, samples of a resin-based sealant containing varying amounts of SB, a zwitterionic additive, were evaluated and compared to an unmodified sealant. The results indicate that the null hypotheses can be rejected due to the observation of significant differences in the mechanical properties and bacterial resistance between the control and the SB-containing samples.

The approach of incorporating SB to maintain a balance of negative and positive charges and reduce the adhesion of serous protein and biomolecules has been extensively used in developing surface coatings for biomedical devices and implants [18]. However, reports have stated that the mode of action of SB in dental materials is different than that in other applications due to the complex interactions inherent to a dynamic oral environment [10]. The complexity of the oral environment stems from its continuous interaction with the external environment, which results in multi-biome habitat formations. This predisposes the oral environment to pathologies like dental caries. Furthermore, *S. mutans* has been widely identified as one of the leading causative microorganisms for dental decay. Thus, the focus of the present research was to supplement the physical barrier mechanism of resin sealant with anti-biofouling properties. In the current study, resin-based sealants containing varying concentrations of SB were tested to evaluate their mechanical properties and the effect of SB incorporation on protein and bacterial adhesion.

### 3.1. Mechanical Properties

As seen in Figure 1A, increasing the level of SB incorporation decreased the flexural strength. While the difference between the control (74.4 ± 12.2 MPa) and the 1.5% SB samples (66.4 ± 8.0 MPa) was not statistically significant, a statistically significant reduction in the flexural strength relative to the control was observed with the addition of 3% SB (53.4 ± 12.9 MPa) and 5% SB (36.8 ± 8.4 MPa).

The results of elastic modulus testing were similar to the flexural strength testing results. There was no significant difference between the control (2748.4 ± 315.4 MPa) group, the 1.5% (2784 ± 711.3 MPa), and the 3% (2202.6 ± 288.3 MPa) SB groups. However, the elastic modulus of the 5% SB group (1496.14 ± 417.1 MPa) was significantly lower than those of the control and the 1.5% SB group (Figure 1B).

### 3.2. Wettability

The results of wettability tests are shown in Figure 1C. No significant difference between the water contact angles of the control (64.06° ± 3.11°) and the 1.5% SB samples (61.48° ± 3.04°) was observed. However, significant reductions compared to the control were observed in the 3% (57.74° ± 2.03°) and 5% (56.54° ± 3.30°) SB groups.

### 3.3. Depth of Cure

The results of the depth of cure testing are shown in Figure 1D. The mean values for the control samples, 1.5% SB, 3% SB, and 5% SB, were 7.53 ± 0.25 mm, 7.76 ± 1.08 mm, 7.55 ± 0.38 mm, 7.57 ± 0.34 mm, respectively. There were no significant differences in the depth of cure among the groups, indicating that the incorporation of SB did not affect the polymerization of the sealant.

Regarding mechanical properties, significant reductions in the strength and modulus of elasticity were observed upon incorporation of 3% or 5% SB in the resin. Although incorporating 1.5% SB resulted in an approximately ten percent reduction in the flexural strength compared to the commercial control, the reduction was determined to not be statistically significant. A similar trend of reduction in the bond strength and flexural modulus upon incorporation of other zwitterionic compounds was observed in previous studies [19,20,21,22,23]. The possible reason for this result is associated with the heterogeneity of SB in resin-based sealants, which results in a degree of exclusion from the interface and clusterization of SB in the polymer matrix [23]. In addition, this change in the flexural strength may occur as the SB content increases, and so the mixing and distribution of SB becomes more difficult, which results in a decreased resistance of the material [11]. The changes in strength have been attributed to the possible clustering of quaternary ammonium species (QAS) in the resin matrix, which alters the internal stress dissipation. In addition, the hydrophilic nature of the zwitterionic molecules can potentially cause an increase in the plasticization of the material, interfering in the polymer chain formation. Adding constituents could also potentially affect the curing performance. This is particularly significant for class 2 sealants, where an external energy source is vital for polymerization (ISO 6874:2005). The principal use of sealants is to block pits and fissures of varying depths on the enamel surface, making good curing depth a critical feature. Therefore, the depth of the cure of the modified sealants was evaluated as per the ISO guidelines. The depth of cure for SB-modified samples was observed to be comparable to that of the commercial control, so any potential adverse effects of SB incorporation on the curing potential and depth of cure could be excluded. The changes in surface wettability can be explained in part by the hydrophilic nature of zwitterions. A surface’s wettability is primarily determined by its surface properties. In this work, the surfaces were modified by the incorporation of zwitterionic moieties into the polymers. These polymers are believed to have low interfacial energy with water, resulting in a reduction in the contact angle with water [24]. This trend was observed in the present study: there was a continuous decrease in the contact angle with the increasing incorporation of SB. The role of SB or other zwitterions in affecting the wettability is further supported by the observation that the surfactant activity of zwitterionic groups in polymers is reduced due to an increased entropy demand after polymerization [9].

### 3.4. Protein Adsorption

The amounts of protein adsorbed from the BHI medium on the samples with 3% SB and 5% SB were significantly lower than the amount adsorbed on the control (Figure 2A). Figure 2B indicates that the amounts of adsorbed BSA follow the same trend: the 3% SB sample showed a significantly lower adsorption compared to those of the control and 1.5% SB samples, and there was no significant difference in the adsorption between the 3% SB and 5% SB samples.

Another essential property for an active caries-protective material is its ability to withstand salivary protein adsorption. Such adsorption results in the formation of salivary pellicles, which allows for the attachment of primary colonizers [25,26]. The marked reduction in BSA adsorption and the adsorption of proteins from the BHI medium with the increasing SB content observed in this study suggests SB-incorporated resin-based sealants can prevent the formation of such proteinaceous pellicles. Similar findings were reported in earlier studies that described the incorporation of other QAS such as dimethylaminohexadecyl methacrylate [15,23], methacryloyloxyethyl phosphorylcholine [10], and octafluoropentyl methacrylate [27]. The influence of sulfobetaine in particular has also been substantiated in earlier studies analyzing surface coatings [14,28,29], including dental varnish [10].

### 3.5. Bacterial Attachment

The representative bacterial viability staining results are shown in Figure 3. As seen in Figure 3A, the surface of the control group showed more bacteria stained with green fluorescence compared to the surfaces of the test groups, indicating the test groups had fewer live bacteria. Although not dramatic when compared to the results shown in Figure 3A, it is confirmed that these visual results were similar to the quantitively CFU counts, as seen in Figure 3B; the number of colonies on the 1.5% SB, 3% SB, and 5% SB samples were reduced significantly compared to the number of colonies on the control sample. In addition, it was observed that almost no red fluorescence was found in this study. It can be seen that SB also causes a decrease in the bacterial attachment to the sealant surface due to the antifouling effect rather than the bactericidal effect, as shown in previous studies of dental biomaterials containing zwitterions such as MPC [8,10,11].

In addition to evaluating resistance to non-specific protein adsorption, the present study also evaluated the specific response to *S. mutans* interaction. According to Bowen and Koo [30], the activity of *S. mutans* is not solely based on pellicle formation; it is also directly influenced by locally-produced glucans, which alter surface topographies and provide specific binding sites [31]. The evaluation of cell viability showed a marked reduction in the number of *S. mutans* CFUs, both quantitatively and qualitatively (i.e., visually). This finding is particularly promising for increasing the efficacy of sealant in caries prevention. Similar findings were also reported for the incorporation of other QAS, including marked reductions in demineralization [10,32] and white spot lesion formation [33].

### 3.6. Cell Cytotoxicity

The results of cell cytotoxicity tests are shown in Figure 4. The positive control sample had a significantly higher cytotoxicity than any other group. There were no statistically significant differences in the cell cytotoxicity between the negative control, control, 1.5% SB, 3% SB, and 5% SB samples.

Campos et al. reported a significant improvement in anti-bacterial performance against *S. mutans* in a recent study. However, the cytotoxicity of the modified polymer was adversely affected [23]. In this study, cytotoxicity was examined in accordance with ISO 10993. MTT tests give an objective assessment of the reduction in cell viability; for the work described here, a quantitative assessment was performed using L929 cells. In contrast to the findings of Campos et al., the addition of SB did not produce any adverse cytotoxicity outcome; results were similar to the control group. It should also be noted that an earlier study with low concentrations of different QAS showed acceptable results by MTT analysis compared to methyl methacrylate monomer, which is commonly used in dentistry [34]. Therefore, through careful consideration and selection, an appropriate QAS, such as SB, can be used in combination with resin-based sealants.

This study was performed in an in vitro setting, which is distinctively different from the complex oral microbiome environment. However, promising results with potential practical significance were observed. With the present study design, a 1.5–3 wt% SB incorporation produced positive outcomes in both mechanical and biological tests. There is a limitation in regard to the durability, as all findings may be limited to a short period of time without any longer application. In this study, leaching-related experiments were not conducted. If SB leaches over time, the anti-biofouling effect of SB will not last long, and a bactericidal effect may also occur depending on the amount of leaching. However, according to our previous study using poly(methyl methacrylate), these zwitterions such as MPC and carboxybetaine methacrylate were successfully conjugated to the base material and successfully inhibited the adhesion of human salivary biofilm even when stored in distilled water for seven days [35].

In light of these limitations, future projects aimed at analyzing the longevity, mechanism, and quantitatively in-vivo response using the human salivary biofilm model to the use of SB-incorporated pit and fissure resin-based sealants can be planned.

## 4. Conclusions

From the results of this study, it can be concluded that the incorporation of the zwitterionic compound SB into resin-based sealants improves resistance to protein and bacterial adhesion. The addition of 1.5–3 wt% of SB shows the most balanced beneficial effects with no observable adverse effects. Resin-based sealants containing SB are thus expected to potentially play a significant role in both the prevention and progression of dental caries.

## Figures and Tables

**Figure 1 polymers-12-01716-f001:**
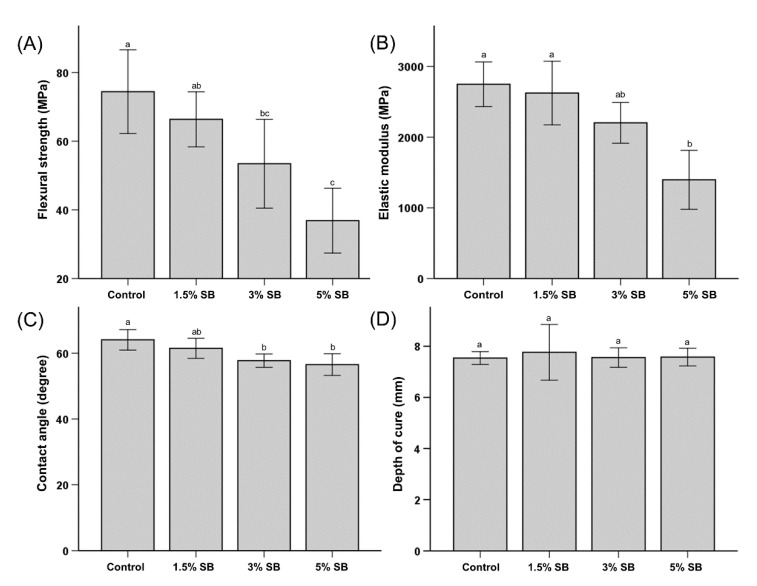
Comparison of the flexural strength (**A**), elastic modulus (**B**), water contact angle (**C**), and depth of cure (**D**) among different groups of samples. Different lowercase alphabetical letters above the bars indicate significant differences.

**Figure 2 polymers-12-01716-f002:**
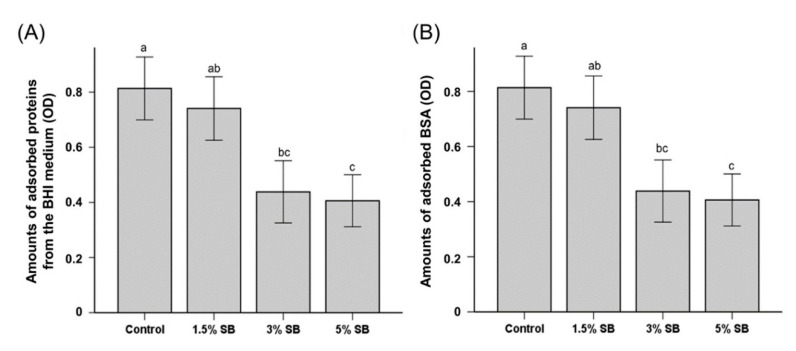
Comparison of the optical density (OD) of the protein adsorbed from a brain heart infusion (BHI) medium (**A**) and adsorbed bovine serum albumin (BSA (**B**)) among resin sealants with different concentrations of sulfobetaine methacrylate (SB). The OD is proportional to the amount of adsorbed protein. Different lowercase alphabetical letters above bars indicate significant differences.

**Figure 3 polymers-12-01716-f003:**
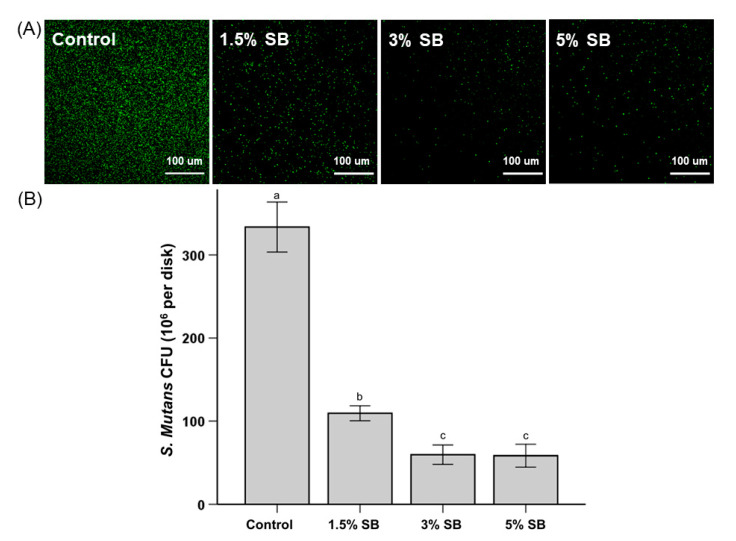
Representative live/dead staining images of bacteria (*S. mutans*) attached on the surfaces of a control sample and samples of SB-modified sealants with varying wt% of SB (**A**). Colony-forming unit (CFU) counts of *S. mutans* attached on the surfaces with different concentrations of SB (**B**). The different lowercase alphabetical letters above the bars indicate significant differences.

**Figure 4 polymers-12-01716-f004:**
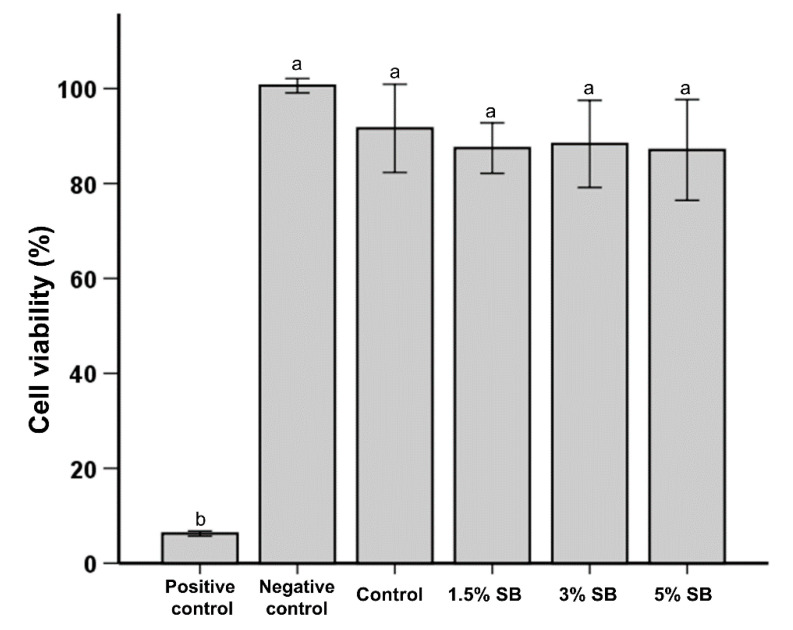
Comparison of the L929 cell viability among control samples and samples with different percentages of SB. The same lowercase alphabetical letters above the bar graph indicate there are no significant differences between the groups.

**Table 1 polymers-12-01716-t001:** Composition of materials in the control and experimental groups.

Groups	Group Code	Sealant (wt%)	SB (wt%)
1	Control	100	0
2	1.5% SB	98.5	1.5
3	3% SB	97	3
4	5% SB	95	5

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
