# Peer review of "Improvement in the Microbial Resistance of Resin-Based Dental Sealant by Sulfobetaine Methacrylate Incorporation"

_polymers, 2020, doi:10.3390/polym12081716_

Round 1

Reviewer 1 Report

This manuscript presents interesting results regarding the incorporation of the bioactive additive sulfobetaine methacrylate into a commercially available resin-based dental sealant at different ratios. The authors evaluated relevant physical-chemical properties of the blends demonstrating that the wettability and the curing properties of the resin were not affected by the additive. A change in the mechanical properties is observed in the presence of higher concentrations of the additive, with reasonable explanation of these effects in terms of clusterisation and the hydrophilic nature of the additive. However, there does not seem to have a proportional effect on the contact angle values and this is not explained. The authors evaluated protein adsorption on the materials, showing a significant effect in the presence of the additive that also correlates with a reduction in the number of bacterial cells attached to the surface. The modified materials show also good biocompatibility tested on model fibroblast cells. The work is presented in a clear style and most of the results and conclusions are supported by experimental evidence. It is recommend that this work can be accepted for publication after the issues listed below are addressed.

  1. The authors claimed that there is a significant difference within contact angle data, yet variations between the average values of different samples are small. This is in clear contrast with the mechanical properties that changed dramatically. The combination of these results may suggest clusterisation of the additive within the polymeric matrix, and a degree of exclusion from the interface. The authors should discuss this in the context of existing literature on polymer blends.
  2. In Fig3A only Syto9 (Live bacteria) data is shown, showing a reduction on the number of cells. LIVE/DEAD data is generally presented showing the two separated channels (Green for Syto9 and Red for PI) to differentiate between the reduction of the total number of attached cells and the killing effect. The authors are encouraged to present the two separate channels for live and dead cells and discuss the data accordingly. Can the authors attempt quantification of biofilm coverage using fluorescence microscopy data?
  3. Live/Dead imaging in Fig3A showed a dramatic reduction of the number of live cells, but the CFUs showed less than an order of magnitude difference. Could the authors explain this apparent discrepancy?
  4. No data on leaching of the additive is presented. This is particularly relevant for the final application and on the mechanism of action, i.e. release of the active could lead to biocidal effects on planktonic bacteria or eventually to deactivation of the material by depletion of the additive from the surface. Did the authors attempted evaluation of the effect on planktonic bacteria or any other experiment to probe  leaching of the active additive?

Author Response

Thank you for your thoughtful comments that have helped improve the quality of our manuscript. We agree that manuscript would need more explanations of cluterisation and hydrophilic nature of the zwitterionic additive. In accordance with your comments, we have modified this manuscript.

Comment 1:

Thank you for your valuable comments. We agree that despite improved antibacterial effect with SBMA, additive would result in deterioration of mechanical properties. According to your suggestions, the limitations of present study was modified in “Discussion” section as follows:

The possible reason for this result is associated with the heterogeneity of SB in resin-based sealant, which results in a degree of exclusion from the interface and clusterisation of SB in the polymer matrix [23]. In addition, this change in flexural strength may occur as the SB content increases, mixing and distribution of SB becomes more difficult, which result in decreasing the resistance of the material [11]

Comment 2:

The images of LIVE/DEAD staining showed that there were no red coloured bacteria on any of samples for Streptococcus mutans and therefore no red found on the image. Although there was no biofilm data in our study, previous studies have confirmed the anti-biofouling effect of multi-species biofilm through biofilm thickness and biofilm mass. As reviewer said, additional fluorescence microscopy data was needed using biofilm. Therefore, we described in the “Discussion” section that the antimicrobial effect of SB should be demonstrated through biofilm data. According to your comments, the contents are also reflected in the “Results and Discussions” section as follows:

In addition, it was observed that almost no red fluorescence was found in this study. It can be seen that SBMA also causes a decrease in the bacterial attachment to the sealant surface due to the antifouling effect rather than the bactericidal effect, as shown in previous studies of dental biomaterials containing zwitterions such as MPC [8, 10, 11].

There is a limitation in regard to the durability, all findings may be limited to a short period of time without any longer application. In this study, leaching-related experiments were not conducted. If SB leaches over time, the anti-biofouling effect by SB will not last long, and a bactericidal effect may also occur depending on the amount of leaching. However, according to our previous study using poly(methyl methacrylate), these zwitterions such as MPC and carboxybetaine methacrylate were successfully conjugated to the base material and successfully inhibited the adhesion of human salivary biofilm even when stored in distilled water for 7 days [35].

In light of these limitations, future projects aimed at analyzing longevity, mechanism, and quantitatively in-vivo response using the human salivary biofilm model to the use of SB-incorporated pit and fissure resin-based sealants can be planned.

Comment 3: 

We apologize for the confusion caused by lack of expression. While LIVE/DEAD image is a representative image, CFUs are a number representing the average values of repeated experiments. Although it may seem inconsistent, the overall trend is considered to be similar. We have rewritten the relevant sentences as follows:

Representative bacterial viability staining results are shown in Figure 3. As seen in Figure 3A, the surface of the control group showed more bacteria stained with green fluorescence compared to the surfaces of the test groups, indicating the test groups had fewer live bacteria. Although not dramatic when compared to Figure 3A results, it is confirmed that these visual results were similar to the quantitively CFU counts, as seen in Figure 3B; the number of colonies on the 1.5% SB, 3% SB, and 5% SB samples were reduced significantly compared to the number of colonies on the control sample.

Comment 4:

Thank you for your helpful comments. We fully agree with your opinion that it needs a data on leaching of the additive. To overcome these limitations, further studies are needed to consider the effect on bacteria and leaching the additive. According to your comments, the contents are also reflected in the “Results and Discussions” section as follows:

There is a limitation in regard to the durability, all findings may be limited to a short period of time without any longer application. In this study, leaching-related experiments were not conducted. If SB leaches over time, the anti-biofouling effect by SB will not last long, and a bactericidal effect may also occur depending on the amount of leaching. However, according to our previous study using poly(methyl methacrylate), these zwitterions such as MPC and carboxybetaine methacrylate were successfully conjugated to the base material and successfully inhibited the adhesion of human salivary biofilm even when stored in distilled water for 7 days [35].

In light of these limitations, future projects aimed at analyzing longevity, mechanism, and quantitatively in-vivo response using the human salivary biofilm model to the use of SB-incorporated pit and fissure resin-based sealants can be planned.

Reviewer 2 Report

It is an interesting topic and suitable to be published in Polymers. However, I do have some suggestion and comment to the author before accepting.

  1. Intro., could the author mentioned what the other materials were reported to prevent dental caries.
  2. I suggest the author mention more detail of why a zwitterionic is an ideal non-fouling material in Intro 
  3. The last paragraph of the Intro needs to describe more details about the experiment design.
  4. Did the author try to incorporate more than 5% SB? could it enhance more anti-bacteria?
  5. Does different pH affect the bacteria or protein attach performance 

Author Response

Comment 1:

Thank you for your helpful comments. According to your comments, we have added the relevant sentence as follows:

Various materials such as toothpaste, mouth rinse and fluoride varnish have been used to prevent dental caries. Since the pits and fissures of occlusal surface have complicated and irregular structures, it is difficult to prevent dental caries using these materials [1].

Comment 2:

Taking your suggestion, the “Introduction” is modified to include the details as follows:

On the surface of the dental material containing zwitterion, a thick hydration layer is formed to block the adhesion of the salivary protein, thereby preventing oral bacteria and fungi from attaching to the surface of the material [10,11]. In particular, the synergetic effect between bioactive components of resin-based material was evident with increased hydrophilicity by the addition of zwitterion. The previous study showed antimicrobial properties as well as acid neutralizing capacities due to the zwitterionic nature of the components [11].

Comment 3:

Thank you once again for your valuable comments. According to your recommendation, we have revised the manuscript as follows:

Accordingly, the aim of this study was to evaluate the mechanical, antimicrobial activity, and cytotoxicity of low percentages of SB-incorporated resin-based sealant. The null hypothesis was that the incorporation of SB will have no significant effect on the mechanical properties, bacterial resistance, or cytotoxicity of the resin-based sealant.

Comment 4: 

Thank you very much for your comments and we do apologize for lack of information. Based on many previous studies, the addition of SB in more than 5% were would result in ineffective physical/mechanical properties of the original materials. Additionally, following sentence has been added as follows:

From the results of preliminary experiments, it was confirmed that the content of SB of 5 wt% or more could dramatically decrease the mechanical properties of the base material, and based on this, the composition of the test group was confirmed in this study. Four sealant compositions were tested, as listed in Table 1. In each experiment, at least five specimens were tested for each group.

Comment 5:

The relevant information has been added as follows:

Generally, acid produced by carbohydrate-fermenting acidogenic bacteria and demineralization dominate when the pH is below the critical pH of 5.5, which results in the dissolution of the enamel minerals. According to the previous study [10,11], The hydrophilicity of zwitterions allowed water to easily reach the core of the base material, which increases the ion release of the bioactive dental material, which promotes the acid-neutralizing effect.

Reviewer 3 Report

This is an interesting manuscript on the modification of resin-based sealant to improve the microbial resistance. The work is original and relates the interdisciplinary knowledge. The are just few small comments/suggestions:
1. Introduction - it is suggested to extend this part (basing on the statements in the lines 46-48 and 50-52).
2. Lines 76-77 - something is missing in that sentence "For each group in Table 1, five 25 mm × 25 mm × 2 mm samples were."
3. Lines 148-150 - the description of the statistical analysis should be extended.
4. What was the accuracy of measurements?
5. What is the chemical compatibility between SB and the resin-based dental sealant? 
5. It is suggested to combine the results with the discussion sections into one "Results & Discussion" section - it would be much easier to follow the experimental results and their interpretation.
6. What is the durability of such novel sealants?

Author Response

Comment 1:

Thank you for your helpful comments. According to your comments, “Introduction” section has been modified as follows:

Various materials such as tooth paste, mouth rinse and fluoride varnish have been used to prevent dental caries. Since the pits and fissures of occlusal surface have complicated and irregular structures, it is difficult to prevent dental caries using these materials [1].

On the surface of the dental material containing zwitterion, a thick hydration layer is formed to block the adhesion of the salivary protein, thereby preventing oral bacteria and fungi from attaching to the surface of the material [10,11]. In particular, the synergetic effect between bioactive components of resin-based material was evident with increased hydrophilicity by the addition of zwitterion. The previous study showed antimicrobial properties as well as acid neutralizing capacities due to the zwitterionic nature of the components [11].

Generally, acid produced by carbohydrate-fermenting acidogenic bacteria and demineralization dominate when the pH is below the critical pH of 5.5, which results in the dissolution of the enamel minerals. According to the previous study [10,11], The hydrophilicity of zwitterions allowed water to easily reach the core of the base material, which increases the ion release of the bioactive dental material, which promotes the acid-neutralizing effect.

Accordingly, the aim of this study was to evaluate the mechanical, antimicrobial activity, and cytotoxicity of low percentages of SB-incorporated resin-based sealant. The null hypothesis was that the incorporation of SB will have no significant effect on the mechanical properties, bacterial resistance, or cytotoxicity of the resin-based sealant.

Comment 2:

According to your suggestion, we’ve revised the manuscript as follows:

For each group in Table 1, the bar-shaped samples (25 mm × 25 mm × 2 mm) were prepared.

Comment 3:

According to your suggestion, we’ve supplemented the manuscript on the statistical analysis as follows:

All statistical analyses were performed using IBM SPSS soft-ware, version 23.0 (IBM Korea Inc., Seoul, Korea) for Windows. The data are expressed as the mean ± S.D. of at least three independent experiments. Statistical significance was evaluated by one-way analysis of variance with Tukey’s post hoc test. Values of p < 0.05 were considered statistically significant.

Comment 4:

All the samples were prepared based on the previous studies and ISO standards. To increase the accuracy of the measurements, at least five specimens were tested for each group in each experiment. The data are expressed as the mean ± S.D. of at least three independent experiments.

Comment 5: 

The relevant information has been added in the limitation of this study as follows:

There is a limitation in regard to the durability, all findings may be limited to a short period of time without any longer application. In this study, leaching-related experiments were not conducted. If SB leaches over time, the anti-biofouling effect by SB will not last long, and a bactericidal effect may also occur depending on the amount of leaching. However, according to our previous study using poly(methyl methacrylate), these zwitterions such as MPC and carboxybetaine methacrylate were successfully conjugated to the base material and successfully inhibited the adhesion of human salivary biofilm even when stored in distilled water for 7 days [35].

Comment 6:

Thank you very much for your specific comments. According to your suggestion, we have revised the manuscript from each “Results”, “Discussions” section to “Results and Discussion” section.

Comment 7:

Thank you once again for your valuable comments. The relevant information has been added in the limitation of this study as follows:

There is a limitation in regard to the durability, all findings may be limited to a short period of time without any longer application. In this study, leaching-related experiments were not conducted. If SB leaches over time, the anti-biofouling effect by SB will not last long, and a bactericidal effect may also occur depending on the amount of leaching. However, according to our previous study using poly(methyl methacrylate), these zwitterions such as MPC and carboxybetaine methacrylate were successfully conjugated to the base material and successfully inhibited the adhesion of human salivary biofilm even when stored in distilled water for 7 days [35].
